# How self-efficacy beliefs in dealing with negative emotions are associated to negative affect and to life satisfaction across gender and age

Mariagiovanna Caprara[1]*, Laura Di Giunta[2], José Bermúdez[1], Gian Vittorio Caprara[2]

**1** Department of Personality Psychology, Evaluation and Psychological Treatment, National University for Distance Education (UNED), Madrid, Spain, **2** Department of Psychology, University of Rome "La Sapienza", Rome, Italy

* gcaprara@psi.uned.es

**Data Availability Statement:** All relevant data are within the manuscript and its Supporting Information files.

## Abstract

The present study examines the extent to which individuals' self-efficacy beliefs about their capacity to manage distinct emotions, such as anger, sadness, fear, shame and guilt, are associated with negative affect and life satisfaction in a Spanish population of diverse ages. The results attest to the validity of the Multidimensional Negative Emotions Self-Regulatory Efficacy Scale (MNESRES) and show that self-efficacy beliefs related to basic and self-conscious/moral emotions are associated differently with negative affect and life satisfaction. These findings corroborate previous findings from American and Italian populations, and they support the view that discrete emotions deserve distinct attention, either regarding their management or their association with individuals' well-being and adjustment.

## Introduction

Social cognitive theorists view personality as a cognitive-affective system whose functioning depends upon the concerted action of psychological structures that have gradually been established to enable individuals to exert some control over their experiences, and the course of their lives [1]. Self-efficacy beliefs have been assigned a special role in attesting to human agency's distinct capacities for self-reflection and self-regulation, and in promoting successful adaptation [2]. A large body of research reflects how the beliefs people hold about their capacity to cope efficaciously with arduous challenges and to face demanding situations, exert a pervasive influence over thought, motivation and action across different domains of functioning [1]. Originally, self-efficacy beliefs were considered as highly contextualized knowledge structures that affect appraisal, which in turn guide actions [2, 3]. This led researchers to focus on specific tasks and to pursue a multifaceted approach to study the various expressions of self-efficacy in diverse situations. However, it later became evident that the influence of self-efficacy beliefs extends beyond specific skills, tasks and situations, as individuals reflect upon the consequences of their actions and use their experience to broaden their understanding, and to control their feelings and thoughts. This raised the question as to how self-efficacy beliefs might affect individuals' capacity to manage their emotions [4].

**Funding:** The author(s) received no specific funding for this work.

**Competing interests:** The authors have declared that no competing interests exist.

## Regulation of emotions

Research on emotion management has flourished in recent decades, with a special focus on the regulation of emotions [5–7]. This phenomenon covers the expression and management of both positive and negative emotions [8], and adequate regulation has been associated with health and well-being [9, 10]. Individuals differ in how they think about emotions, and their beliefs about the controllability of emotions significantly affect how they acknowledge the need to regulate them by selecting and implementing adequate strategies, and monitoring the outcomes of their endeavours. Indeed, the control that individuals believe they can exert over the causes of their emotions and over the consequences of their reactions strongly influences the regulatory strategies they employ [11, 12]. It is likely that the more people who believe that emotions can be regulated, the more of them who would be willing to manage these emotions. As self-efficacy beliefs have been proven to exert a pervasive influence over thought, motivation and action across different functional domains, individuals' beliefs about the control they can exert on their emotions are thought to be crucial to achieve effective regulation [2, 4, 5].

## Measuring individuals' perceived efficacy in the regulation of basic and self-conscious/moral negative emotions

Recent studies on perceived self-efficacy have broadened to consider how self-efficacy beliefs affect people's capacity to manage their emotions, and their impact on individuals' psychosocial functioning and well-being. Initially, this led to the development of the Regulatory Emotional Self-Efficacy scale (RESE), which has been used widely in several countries [13–15]. The scale conserves the distinction between positive and negative affect, and it includes five items that assess individuals' perceived self-efficacy in expressing positive emotions like joy, enthusiasm and pride, as well as nine items that assess individuals' perceived self-efficacy in managing anger/irritation and despondency/sadness. The RESE scale is becoming increasingly well validated, and there is a growing body of evidence that people's beliefs about their capacity to express positive emotions and to control negative emotions exert a crucial influence on their successful development and social adaptation [14–16].

Most of the earlier research on affective self-regulatory efficacy addressed basic emotions, and their relevance to health and social functioning [13, 17–19]. Focusing on negative emotions has led to the development of the Multidimensional Negative Emotions Self-Regulatory Efficacy Scale (MNESRES) [19], a new measure that includes five subscales designed to assess self-efficacy beliefs in relation to the management of anger/irritation, sadness, fear, shame and guilt (SE_Anger, SE_Sadness, SE_Fear, SE_Shame and SE_Guilt, respectively). In validating MNESRES it was found that five distinct mastery beliefs were uniquely associated with social adjustment and well-being [19]. However, confirmatory factor analyses showed that the scales could be traced to either five correlated factors or to two second-order factors, which in turn could be traced to a third-order factor. Specifically, while SE_Anger, SE_Sadness and SE_Fear could be traced to a common second-order factor that may be considered to be related to basic negative emotions, SE_Shame and SE_Guilt could be traced to another second-order factor thought to be related to self-conscious moral emotions. Both models fitted reasonably well with the data, but the one that best fitted the data in Italy and the U.S. was a third model, one that posited a third-order factor above (1) a second-order factor that included SE_Anger and SE_Sadness (2), SE_Fear as a first-order factor by itself, and (3) another second-order factor that included SE_Shame and SE_Guilt [19].

## Self-efficacy beliefs related to basic and self-conscious/moral emotions

People display basic emotions like anger, sadness and fear from the early stages of life. By contrast, self-conscious/moral emotions like guilt and shame do not appear until self-structures

and processes develop, enabling individuals to reflect on their experiences, and to acknowledge the discrepancies between their behaviour and their values [20]. The basic emotions anger/irritation, sadness and fear are thought to have evolved over time as part of humans' basic equipment to cope with events such as loss, failure and threat to survival [21, 22]. In general, they occur with moderate engagement of cognitive processes, and they are associated with the triggering of rapid and sometimes automatic responses [23, 24]. The regulation of these emotions has received much attention due to the adverse consequences arising from their dysregulation in terms of health and social integration [25]. Anger has been mostly associated with externalizing behavioural problems [26, 27], whereas sadness and fear have been associated with internalizing problems, and sometimes with externalizing problems as well [28].

To date, fewer studies have explicitly addressed the regulation of self-conscious/moral emotions like shame and guilt, although one should note the growing interest in studying these emotions given their significant influence on social behaviour [29, 30]. Shame and guilt engage advanced cognitive processes, resting upon the development of internal self-referent structures and processes, and largely depending upon social and cultural norms. They reflect individuals' internalization of moral obligations or socially expected conduct, the violation of which carries feelings of discontent, and interferes with self-acceptance and respect [31]. In addition, like basic emotions shame and guilt have been shown to affect mental health and well-being, although to a different degree [32–35].

Whereas each emotion can be viewed as the result of a unique organization of affect, cognition and behaviour under given situations and cultural constraints [36], it is likely that self-efficacy beliefs affect the regulation of both basic and self-conscious/moral emotions, and that they influence health and well-being, albeit to different extents. Indeed, negative basic emotions like anger, fear and sadness have a pervasive impact over individuals' life, and their regulation has a primary influence on adaptation when compared to self-conscious/moral emotions like guilt and shame. In terms of the regulation of basic emotions vs. self-conscious/moral emotions, our reasoning was based on two considerations: 1) Basic emotions are fundamental for the individual's survival and as such, they are largely prewired in our brain, manifesting themselves early in development and with similar features across cultures. By contrast, self-conscious/moral emotions appear later in development, and they rely upon the appearance of cognitive structures that underlie the formation of a self-system to which they are instrumental and that maintains behavior within the limits of social and personal standards. Moreover, these emotions are largely conditional, and they are shaped by social experiences and cultural references [37–39]. 2) Basic negative emotions can be viewed as precursors of self-conscious/moral emotions on which reactions to harm and pleasure are founded. While negative basic emotions may represent survival responses to external harm or danger in the outside world, self-conscious/moral emotions underlie self-respect and they respond to violations of internal oughts and beliefs [40]. It is likely that the proper regulation of basic emotions requires capacities for self-reflection, learning and practice that establish conditions for the subsequent development and management of self-conscious/moral emotions. Thus, one would expect that self-efficacy beliefs related to basic emotions would ultimately be more effective in protecting health and fostering well-being than self-efficacy beliefs related to self-conscious/moral emotions.

## Aims and hypotheses

The goal of this study was to further validate the MNESRES by examining how individuals' beliefs that they are capable of handling negative emotions like anger, sadness, fear, shame and guilt (i.e.: the dimensions of the MNESRES) are related to negative affect and life satisfaction

in an adult Spanish population. The study was carried out on a large sample that included males and females of different ages. The size and variation in the sample allowed the evaluation of the extent to which self-efficacy beliefs in terms of the individual's capacity to manage distinct negative emotions is associated with negative feelings and life satisfaction, across ages and gender.

Based on earlier findings, we first set out to examine the extent to which previous factorial models could be corroborated and then, to ascertain the invariance of the distinct scales across gender and age-groups [19]. In principle, we expected that items designed to assess each scale (*SE_Anger*, *SE_Sadness*, *SE_Fear*, *SE_Shame*, and *SE_Guilt*) would rely on separate but related first-order factors. Yet we wondered whether the model positing five oblique factors would fit the data better than alternative models tracing beliefs of mastering negative emotions to a hierarchy of higher-order factors. Likewise, we expected a significant similarity in the factorial structure of the *MNESRES* scale across age and gender. In accordance with previous findings [41], we expected higher values for men's self-efficacy beliefs in managing negative emotions than for women. Indeed, women have shown a lower perceived efficacy in managing negative affect than men [13, 42]. Yet women's perceived efficacy in managing negative emotions has shown to improve with age [41, 42], whereas it has shown to decline in men.

For both men and women, perceived efficacy in managing negative emotions has resulted positively associated to life satisfaction [43] and successful adaptation [37], and negatively associated with depressive feelings, shyness and delinquency [13]. Thus, we expected that perceived self-efficacy in managing negative emotions would result in a negative association with negative affect and in a positive association with life satisfaction. However, in accordance with the aforementioned considerations regarding the impact that basic emotions have on social adjustment and well-being, we expected that self-efficacy beliefs related to the management of basic emotions like anger, sadness and fear would be more strongly associated with both negative affect and life satisfaction than self-efficacy beliefs related to the management of self-conscious/moral emotions.

## Materials and methods

### Participants and procedures

Data were collected between 2014 and 2017 from a total of 1695 participants (49% males), who ranged from 19 to 94 years of age (M = 43.34, SD = 13.48: see Table 1). The participants were recruited at the Open University in Madrid (UDIMA), and from the relatives and friends of undergraduate Psychology students who received course credits for recruiting them. All the participants were born in Spain, were considered to belong to the middle class, were moderately-to-highly educated, and the majority of them lived in the Municipal area of Madrid.

Forward-and back-translation was used to ensure the linguistic and conceptual equivalence of the measures across languages [44]. The Spanish versions of the questionnaires were

**Table 1. Distribution of the interaction between age and gender in the Spanish sample.**

| Age group[1] | N Men | N Women | N Total | Mean | SD |
|---|---|---|---|---|---|
| Younger | 390 | 404 | 794 | 32.80 | 5.04 |
| Middle | 428 | 246 | 674 | 46.42 | 4.89 |
| Older | 97 | 130 | 227 | 68.87 | 8.44 |
| Total | 915 | 780 | 1695 | 43.34 | 13.48 |

[1]: Younger = 20–40 years old; Middle = 41–60 years old; Older 61–94 years old.

administered individually to all the participants after obtaining their informed consent. The participants were informed that their responses would be associated with number codes by the research staff and that full confidentiality would be guaranteed throughout all stages of the study. The study and the procedures involved were approved by the Institutional Ethical Research Review Board of the Open University in Madrid (UDIMA), and the data were analysed anonymously.

## Measurement scales

**Multidimensional Negative Emotions Self-Regulatory Efficacy Scale (MNESRES)** [19]. Participants rated their capacity to manage different emotions on a scale of 1–5 (1 = *not at all well*; 5 = *very well*) using 3 items for each: anger/irritation (e.g., "How well can you avoid flying off the handle when you get angry?"), despondency/sadness (e.g., "How well can you keep from getting dejected when you are lonely?"), shame/embarrassment (e.g., "How well can you overcome shame when your weaknesses become evident in front of others?"), fear (e.g., "How well can you stay calm in situations in which many others would be fearful?") and guilt (e.g., "How well can you control feelings of guilt after not fulfilling important commitments and obligations?"). The formulation, selection and phrasing of the challenges presented in each item, carrying different degrees of difficulty, followed Bandura's guidelines [45].

**Negative affect.** Participants rated the degree to which they generally experience the ten negative emotional states in the Positive and Negative Affect Scale (PANAS) [46], on a scale of 1–5 (1 = *not at all*; 5 = *very much*), with a Cronbach's alpha coefficient (α) = .83 for negative affect. For the purposes of this research, we used the Spanish version of this scale [47], and the emotions referred to as "distressed", "upset", afraid", "scared", "nervous", "jittery", "hostile", "guilty", "irritable" and "ashamed" were the ten items included in this score.

**Life satisfaction.** Life satisfaction was measured using the 5 items of the Satisfaction with Life Scale (SWLS) [48]. The participants rated the extent to which they felt generally satisfied with life (e.g. "I am satisfied with my life") using a 7 point Likert scale ranging from 1 (= *strongly disagree)* to 7 (= *strongly agree*), and with α = .89. In this study we used the Spanish version of SWLS [49].

## Analytic approach

In order to investigate the factorial structure of the *MNESRES*, in accordance with previous findings [19], we tested four Confirmatory Factor Models (*CFA*) using a maximum likelihood estimation of the parameters. Model 1, the five types of self-efficacy were considered as separate constructs correlated on the first level. In Model 2, the five types of self-efficacy were modelled as expressions of a second-order latent factor reflecting a general capability to manage negative emotions (SE_Negative Affect). In Model 3, SE_Anger, SE_Sadness and SE_Fear were modelled as expressions of a second-order latent factor reflecting the capability to manage basic negative emotions, whereas SE_Shame and SE_Guilt were modelled as expressions of a different latent second-order factor reflecting the capability to manage self-conscious/ moral emotions. In Model 4, SE_Anger and SE_Sadness were grouped as expressions of a second-order latent factor, SE_Fear was left by itself, and SE_Shame and SE_Guilt were grouped together as another second-order factor. In both Model 3 and Model 4, we also included a latent third-order factor of SE_Negative Affect on which the second-order factors loaded. These models were the most competitive of those tested in a previous study, albeit not in a Spanish context [19]. We focused on the indices of the model's fitness that were least sensitive to sample size, given that obtaining a non-significant $\chi^2$ value becomes increasingly unlikely with larger sample sizes [50]. For each model we calculated the Comparative Fit Index (*CFI*),

the Root-Mean-Square Error of Approximation (*RMSEA*) with its associated Confidence Interval and *p* value, the Standardized Root-Mean-Square Residual (*SRMR*), and the Akaike Information Criterion (*AIC*), a useful index when comparing models that are not nested [51], accepting *CFI* values greater than .95 [46], *RMSEA* values lower than .07 [52] and *SRMR* values lower than .08 [46]. For the *RMSEA*, a non-significant p-value means the model we hypothesized approximates well to the population. Mplus 7 [53] was used for *CFAs*.

To test both the gender and age invariance of the *MNESRES*, we adopted a model-fitting process as indicated elsewhere [54]. We tested three models to examine the gender invariance and three models to test age invariance: configural invariance, the same pattern of fixed and free factor loadings was specified across gender and age-groups, respectively; metric invariance, the same factor loadings for items were specified across gender and age-groups; and scalar invariance, the same intercepts of like item regressions on the latent variables were specified across gender and age-groups. The most frequent additional tests performed were typically those of partial invariance at each step and modification indices for each step were used to refine the structure models [55]. Each model of invariance was nested in the previous model and involved added constraints that built upon previous constraints. Specifically, the model for metric invariance was nested within that for configural invariance and the model for scalar invariance was nested within the one for metric invariance. Difference in the Chi-squared tests were assessed to compare the nested models, adopting a cut-off of $p < .01$ given that obtaining a significant $\chi^2$ value becomes increasingly likely with larger sample sizes [46]. To explore gender and age differences in the dimensions of the *MNESRES*, we examined gender (male and female) by age (younger, middle and older adulthood) with *MANOVA*, using the dimensions of the *MNESRES* as dependent variables. Finally, in order to explore the construct validity of the *MNESRES*, we examined the correlations among the dimensions of *MNESRES*, negative affect and life satisfaction. Moreover, having verified the significant associations among the variables, we conducted two hierarchical regression analyses to examine the associations of the dimensions of the *MNESRES* with negative affect and life satisfaction, respectively. Specifically, the principal effects included age and gender (Step 1), all the dimensions of *MNESRES* (Step 2), and the interaction of age or gender for each dimension of *MNESRES* (Step 3). Lower-order and interactive terms were mean-centred in order to facilitate a correct interpretation of the lower-order terms and to decrease non-essential multicollinearity [56]. MANOVA, correlation and regression analyses were performed with SPSS 18.0 software (SPSS Inc.).

## Results

### Confirmatory factor analyses and reliabilities

The different indices of fit were collected for the four models evaluated: Model 1 $\chi^2(80)$ = 439,196 ($p <001$), *CFI* = .94, *RMSEA* = .05 (.05 - .06), *SRMR* = .04, *AIC* = 67413.04; Model 2 $\chi^2(85)$ = 576,814 (p < .001), CFI = .91, RMSEA = .06 (.05 - .06), SRMR = .05, AIC = 67540.66; Model 3 $\chi^2(86)$ = 514,599 (p < .001), CFI = .92, RMSEA = .05 (.05 - .06), SRMR = .04, AIC = 67476.44; Model 4 $\chi^2(83)$ = 467,502 (p < .001), CFI = .93, RMSEA = .05 (.05 - .06), SRMR = .04, AIC = 67435.35. All models provided a discrete fit to the data according to each index, with Model 1 offering a better fit than the other tested Models according to multiple indices, including the lowest AIC. Although the CFI did not pass the cut-off of .95, it was very close to that threshold, and all the other fit indices were indicative of good fit [35]. Thus, Model 1 appeared to be the model that best represented the structure of the MNESRES in a Spanish context (see Fig 1 for the path diagram of *CFA*). The alpha coefficients for the five sub-scales designed to assess self-efficacy beliefs regarding anger/irritation (SE_Anger),

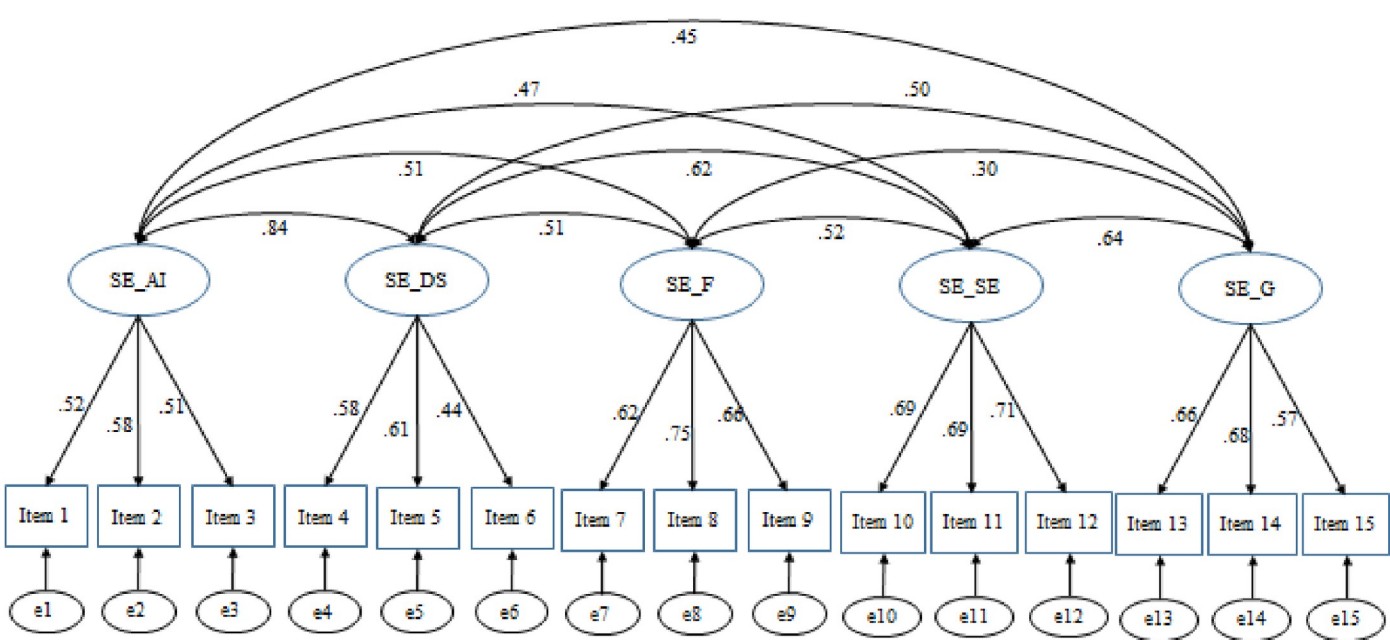

**Fig 1. Path diagram of the best factorial solution of the *MNESRES* in a Spanish context.** The labels in the first order factors refer to the five subscales designed to assess self-efficacy beliefs regarding managing anger/irritation, despondency/sadness, fear, shame/embarrassment, and guilt (*SE_AI, SE_DS, SE_F, SE_SE, SE_G*, respectively). *e1–e15* represent error terms. Standardized factor loadings are shown on the straight arrows. All the paths are significant with p-value < .001.

despondency/sadness (SE_Sadness), fear (SE_Fear), shame/embarrassment (SE_Shame), and guilt (SE_Guilt) management were .55, .54, .71, .74, .67, respectively.

## Gender invariance

Multi-group *CFAs* confirmed that the model with five oblique factors was replicated consistently in men and women. Specifically, the change in the overall chi-squared value between the configural invariance model ($\chi^2$(159, N = 915,780) = 591.14, (p < .001), *CFI* = .92, *RMSEA* = .06 (.05 - .06), *SRMR* = .04) and the metric invariance model ($\chi^2$ (174, N = 915 780 = 608.37 (p < .001), *CFI* = .92, *RMSEA* = .05 (.05 - .06), *SRMR* = .05) was not significant (- $\Delta\chi^2$ (15) = 17.23, p = .31). Likewise, the change in the overall chi-squared value between the metric invariance model and the scalar invariance ($\Delta\chi^2$(10) = 12.74, p = .24) was not significant, suggesting that both the metric and scalar invariance persisted across genders. The fit indices for the scalar invariance were: $\chi^2$(184; N = 915,780) = 621.11, *CFI* = .92, *RMSEA* = .05 (.04 - .06), *SRMR* = .05 (see the first two columns of Table 2 for the unstandardized values of loadings and the intercepts in the five-oblique factor model for men and women). It is important to note that the modification index of all the models evaluated suggested an estimation of covariation between two item errors for males. Our rationale in freely estimating this error covariance was that in the male group, one covariant was estimated freely between item errors 8 (*SE_Fear;* "Overcome feelings of panic and keeping a clear mind in the presence of very dangerous situations") and 11 (*SE_Shame*; "Contain your shame after having made a fool of yourself in front of many people"), both referring to the possibility that events that threaten one's social status can be considered dangerous. This interpretation is in line with a previous study [57] in which feelings of inferiority and threat are fundamental dimensions that underlie behaviour in a social rank system, especially in men. In summary, the test of scalar invariance provided evidence for full scalar variance for all five sub-scales of *MNESRES* between males and females.

**Table 2. Unstandardized values of factor loadings and their intercepts in the five-oblique factor model through gender and age invariance of the MNESRES in a Spanish context.**

| | | Gender Invariance | | Age Invariance | | | | | |
|---|---|---|---|---|---|---|---|---|---|
| | | Factor Loadings | Intercepts | Factor Loadings | | | Intercepts | | |
| | Items | Men-Women | Men-Women | Young | Middle | Late | Young | Middle | Late |
| SE_Anger | 1 | .55 | 3.11 | | .59 | | | 3.11 | |
| | 2 | .59 | 3.09 | | .61 | | | 3.09 | |
| | 3 | .53 | 3.26 | | .54 | | | 3.25 | |
| SE_Sadness | 4 | .56 | 3.29 | | .59 | | | 3.24 | |
| | 5 | .60 | 3.49 | | .60 | | | 3.43 | |
| | 6 | .44 | 3.22 | | .48 | | 3.08 [b] | 3.27 | 3.27 |
| SE_Fear | 7 | .61 | 3.29 | | .60 | | | 3.22 | |
| | 8 | .76 | 3.26 | | .76 | | | 3.17 | |
| | 9 | .64 | 3.39 | .61 | .61 | .74 [a] | 3.30 | 3.30 | 3.42 [b] |
| SE_Shame | 10 | .69 | 3.33 | | .69 | | | 3.26 | |
| | 11 | .70 | 3.04 | | .70 | | | 2.98 | |
| | 12 | .69 | 3.23 | | .69 | | | 3.16 | |
| SE_Guilt | 13 | .65 | 3.09 | .58 [a] | .71 | .71 | 3.00 [b] | 2.99 | 2.99 |
| | 14 | .65 | 2.95 | | .66 | | | 2.87 | |
| | 15 | .57 | 2.81 | .55 | .55 | .73 [a] | 2.72 | 2.72 | 2.74 [b] |

Note. All factor loadings and intercepts were significant at p < .001

[a] factor loading differed significantly from the other two in the same column under *Factor Loadings*

[b] the intercept differed significantly from the other two in the same column under *Intercepts*.

## Age invariance

To examine age invariance in the five subscales of the *MNESRES*, we used the age groups reported in Table 1. There was a significant mean change in the overall $\chi^2$ between the configural invariance model ($\chi^2$ (236, N = 794,674,227) = 589.17 (p < .001), CFI = .94, RMSEA = .05 (.05 - .06), SRMR = .04), and the metric invariance model ($\Delta\chi^2$(28) = 53.25, p = .003), suggesting that full metric invariance does not hold good across the three age-groups. We found that the following equality constraints needed to be lifted for the factor loading of item 13 (*SE_Guilt*; "Control feelings of guilt after not fulfilling important commitments and obligations") in the younger adults, and item 9 (*SE_Fear;* "Stay calm in situations in which many others would be fearful) and item 15 (*SE_Guilt*; "Contain feelings of guilt after having violated very important personal moral principles") in the older adult group. After releasing these constraints on factor loading scores, the mean change in overall $\chi^2$ between the configural invariance model and the partial metric invariance model ($\chi^2$ (261; N = 794,674,227) = 625.29 (p < .001), CFI = .94, RMSEA = .05 (.05–06), SRMR = .06, SRMR = .06) was no longer significant ($\Delta\chi^2$(25) = 36.12, p = .07), suggesting that partial metric invariance held across the three age-groups. We repeated the procedure several times to search for items that were not invariant across the three age-groups based on the modification indices, eventually obtaining a model in which the chi-squared difference between the partial metric invariance model and the partial scalar invariance model was no longer significant ($\Delta\chi^2$(16) = 19.39, *p* = .25). We found that the following equality constraints had to be lifted for the intercepts of item 6 (*SE_Sadness*) and item 13 (*SE_Guilt*) in the younger adults, and intercepts of item 9 (*SE_Fear*) and item 15 (*SE_Guilt*) in the late adulthood group. In summary, the test of scalar invariance provided evidence for full scalar variance for the factors SE_Anger and SE_Shame, whereas it provided

evidence for partial scalar invariance for the SE_Sadness, SE_Fear, and SE_Guilt factors (see the third, fourth, and fifth columns of Table 2 for the unstandardized values of loadings and intercepts in the five-oblique factor model for younger, middle and older adults).

It is important to note that the modification indices estimate the covariation between item errors for some of the groups in all the models evaluated. We estimated five covariations between errors when we had substantive reasons to support such covariation based on the similarities in item content. Our rationale for freely estimating these five error covariances was that one covariation was freely estimated between item errors 4 and 6 in younger adults (both in *SE_Sadness*), these referring to the possibility that being lonely has something in common with being exposed to strong criticism in the young adult phase of life [58]. Covariation between item errors 5 (*SE_Sadness*) and 8 (*SE_Fear*) reflected that both refer to high-risk situations, and that between item errors 8 (*SE_Fear*) and 11 (*SE_Shame*) reflected that the possibility that events which threaten ones social status could be regarded as dangerous situations by the middle adulthood group [59]. Covariation between item errors 1 and 2 (both in *SE_Anger*) refer to a possible attribution bias related to experiencing wrong doing when others give you hard time in late adulthood [60]. Similarly, there was covariation between item errors 8 and 9 (both in *SE_Fear*), which are both about dangerous and fearful scenarios (further details about the content of the items in Spanish are available from the corresponding author.

## Gender and age differences

The overall MANOVA suggested that there are gender [F(5,1681) = 7.5186, p < .001] and age [F(10,3364) = 2.287, p = .01] differences for all five of the *MNESRES* sub-scales, although the gender by age interaction was not significant [F(10,3362) = 1.469, p = .14]. However, the partial $\eta 2$ indicated there were small effects in terms of effect sizes [61] for both the gender (partial $\eta 2$ = .02) and age (partial $\eta 2$ = .01) differences. Given the significant multivariate results, univariate ANOVAs were used to examine each of the *MNESRES* sub-scales (see Table 3), in addition to the estimates of the partial $\eta 2$. Males reported higher self-efficacy beliefs than females when dealing with all the negative emotions examined across all age-groups except for SE_Anger.

## Relationships of the MNESRES with negative affect and life satisfaction

**Correlation.** In order to examine the construct validity of the five *MNESRES* sub-scales, the correlations among the sub-scales were examined, as were the correlations of each sub-

**Table 3. Gender and age differences in the five subscales of the MNESRES in a Spanish context.**

|  | All | | Men | | Women | | Younger | | Middle | | Older | | Gender effects | $\eta^2$ |
|---|---|---|---|---|---|---|---|---|---|---|---|---|---|---|
|  | M | SD | M | SD | M | SD | M | SD | M | SD | M | SD |  |  |
| **SE_Anger** | 3.16 | 0.76 | 3.16 | 0.77 | 3.15 | 0.75 | 3.16 | 0.75 | 3.18 | 0.76 | 3.07 | 0.76 | F(1,1688) = .612, p = .43 | .00 |
| **SE_Sadness** | 3.25 | 0.72 | 3.33[a] | 0.69 | 3.16[b] | 0.75 | 3.25 | 0.70 | 3.26 | 0.71 | 3.26 | 0.82 | F(1,1688) = 16.41, p < .001 | .01 |
| **SE_Fear** | 3.26 | 0.80 | 3.31[a] | 0.80 | 3.20[b] | 0.79 | 3.23 | 0.75 | 3.27 | 0.81 | 3.33 | 0.94 | F(1,1688) = 11.02, p = .001 | .01 |
| **SE_Shame** | 3.14 | 0.81 | 3.20[a] | 0.79 | 3.07[b] | 0.82 | 3.13 | 0.80 | 3.18 | 0.80 | 3.05 | 0.85 | F(1,1688) = 7.30, p = .007 | .01 |
| **SE_Guilt** | 2.89 | 0.76 | 2.95[a] | 0.74 | 2.81[b] | 0.77 | 2.87 | 0.74 | 2.90 | 0.76 | 2.94 | 0.81 | F(1,1688) = 8.70, p = .003 | .01 |

Note. Different letters indicate significant gender differences. None of the univariate age and interactions effects were significant and thus, they were not reported in this Table.

**Table 4. Correlations among the five sub-scales of the MNESRES, negative affect and life satisfaction.**

|  |  | (1) | (2) | (3) | (4) | (5) | (6) | (7) |
|---|---|---|---|---|---|---|---|---|
| **SE_Anger (1)** | r | 1 |  |  |  |  |  |  |
| **SE_Sadness (2)** | r | .465** | 1 |  |  |  |  |  |
| **SE_Fear (3)** | r | .320** | .362** | 1 |  |  |  |  |
| **SE_Shame (4)** | r | .303** | .399** | .389** | 1 |  |  |  |
| **SE_Guilt (5)** | r | .274** | .319** | .214** | .447** | 1 |  |  |
| **Negative Affect (6)** | r | -.192** | -.238** | -.219** | -.129** | -.062* | 1 |  |
| **Life Satisfaction (7)** | r | .194** | .267** | .297** | .179** | .083** | -.295** | 1 |

Note. $p < .01^{**}$.

scale with negative affect and life satisfaction (Table 4). All correlations were significant at $p < .01$ and moderate to strong positive correlations emerged among the five sub-scales. In particular, stronger correlations emerged between SE_Anger and SE_Sadness, and between SE_Shame and SE_Guilt, that is among self-efficacy beliefs related to basic emotions and among self-efficacy beliefs related to self-conscious/moral emotions. Low to moderate negative correlations emerged between all five sub-scales and negative affect and in particular, a stronger correlation was evident between SE_Sadness and negative affect. Low to moderate positive correlations emerged between all five sub-scales and life satisfaction, with a particularly strong correlation between SE_Fear and life satisfaction. The correlation between the two outcomes examined was moderate.

**Regression analyses.** The results of the hierarchical regression analyses focused on negative affect as the dependent variable (Table 5, left). In the first step, age and sex did not account significantly for the variance in *negative affect*, and in the second step, the addition of the five self-efficacy beliefs of *MNESRES* to the regression equation significantly improved the explanation of the variance. However, only self-efficacy beliefs relating to basic emotions (i.e.: SE_Anger, SE_Sadness and SE_Fear) were associated with significant coefficients. In this latter step, self-efficacy beliefs accounted for 8% of the variance in negative affect. In the third step, the subsequent addition of the interaction terms did not account significantly for the variance in negative affect.

The hierarchical regression analyses focusing on *life satisfaction* as the dependent variable were also assessed (Table 5, right) and in the first step, only age significantly and positively predicted life satisfaction. In particular, older people scored higher in life satisfaction than younger ones. In the second step, the addition of the five *MNESRES* self-efficacy beliefs to the regression equation significantly improved the variance explained. However, significant coefficients were only obtained for the self-efficacy beliefs relating to sadness and fear (i.e.: SE_Sadness and SE_Fear), with self-efficacy beliefs accounting for 12% of the variance in life satisfaction in this step. In the third step, the subsequent addition of the interaction terms only moderately accounted for the variance in life satisfaction. In particular, the interaction of SE_Fear and SE_Shame with sex was associated with life satisfaction. We plotted simple slopes of the aforementioned self-efficacy beliefs using high and low values (above and below the median) and accordingly, we found that the association between life satisfaction and SE_Fear was stronger for men than for women (Fig 2A). In addition, the association between life satisfaction and SE_Shame was stronger for women than for men (Fig 2B).

## Discussion

The aim of this study was to examine the degree to which self-efficacy beliefs relating to the management of anger, sadness, fear, shame and guilt are associated with negative affect and

**Table 5. Hierarchical regressions analyses (*HRA*) of sex, age and each of the MNESRES dimensions on negative affect and life satisfaction, respectively.**

| DV: | Negative Affect | | | Life satisfaction | | |
|---|---|---|---|---|---|---|
| First set of *HRA* | **B** [a] | (SE) | $\Delta R^2$ | **B** [a] | (SE) | $\Delta R^2$ |
| Step 1 | | | .003 | | | .01** |
| Sex | .04 | .03 | | .01 | .04 | |
| Age | -.05 | .001 | | **.08**** | .00 | |
| Step 2 | | | **.08**** | | | .12** |
| **SE_Anger** | **-.09**** | .02 | | .05 | .03 | |
| **SE_Sadness** | **-.16**** | .02 | | .17** | .03 | |
| **SE_Fear** | **-.14**** | .02 | | .21** | .03 | |
| **SE_Shame** | -.01 | .02 | | .04 | .03 | |
| **SE_Guilt** | .05 | .02 | | -.05 | .03 | |
| Step 3 | | | .01 | | | **.02**** |
| Sex X age | .001 | .002 | | -.02 | .003 | |
| **SE_Anger X Sex** | .03 | .04 | | -.06 | .06 | |
| **SE_Sadness X Sex** | -.03 | .04 | | .04 | .07 | |
| **SE_Fear X Sex** | .07 | .04 | | **-.12**** | .06 | |
| **SE_Shame X Sex** | .03 | .04 | | **.14**** | .06 | |
| **SE_Guilt X Sex** | -.02 | .04 | | -.09 | .06 | |
| **SE_Aner X Age** | -.06 | .001 | | .05 | .00 | |
| **SE_Sadness X Age** | -.03 | .002 | | .03 | .00 | |
| **SE_Fear X Age** | .05 | .001 | | -.05 | .00 | |
| **SE_Shame X Age** | -.01 | .002 | | -.05 | .00 | |
| **SE_Guilt X Age** | .04 | .001 | | .06 | .00 | |
| Adjusted $R^2$ | .09 | | | .14 | | |
| F value | 9.95** | | | 16.46** | | |

Note.

[a] Standardized regression coefficients.

** $p < .01$. Sex was coded as 0 = men and 1 = women.

life satisfaction, in a population of Spanish adults. To our knowledge, this is the first study conducted on a Spanish population using the MNESRES on a population that breaches at least three generations, young, middle and older adults. CFA showed that the model that best represented the MNESRES structure in accordance with standard criteria is that which proposes five separate but correlated factors. These results differ from previous findings in US and Italian populations of young adults, showing that the best model was that involving a hierarchical structure with a third order latent factor overarching SE_Fear, and two second order factors including SE_Anger and SE_Sadness or SE_Shame and SE_Guilt, respectively. In the present study, we focus on a larger and more diverse sample than that reported previously [19], and the five-oblique-factor solution was that which best fitted the data. Yet one should not forget that alternative models, including that with the best fit previously, were not far from standard criteria of acceptance. In particular, the model that fitted best the data obtained here was one that had previously shown an acceptable fit. Albeit with some caution, it might be concluded that the five-oblique-factor solution is in accordance with the underlying specificity of each emotion, implying that it would be recommendable to focus on each emotion irrespective of its source, manifestations and means of regulation. This recommendation may also be worthy of consideration in terms of self-efficacy beliefs, as corroborated by the five factor solution of the confirmatory factor analysis. However, this does not fully exclude attempting to trace one

*A*

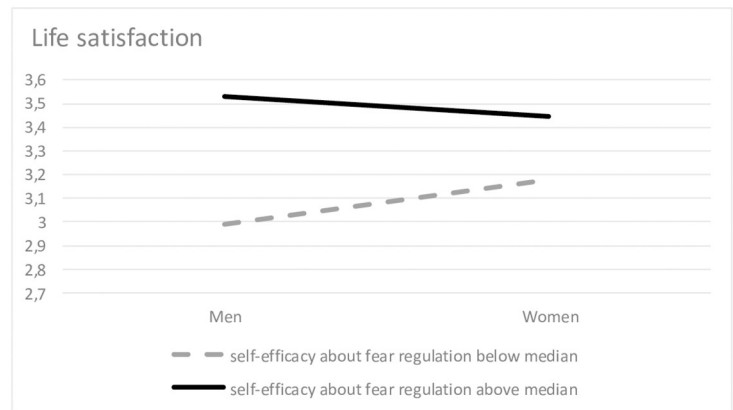

*B*

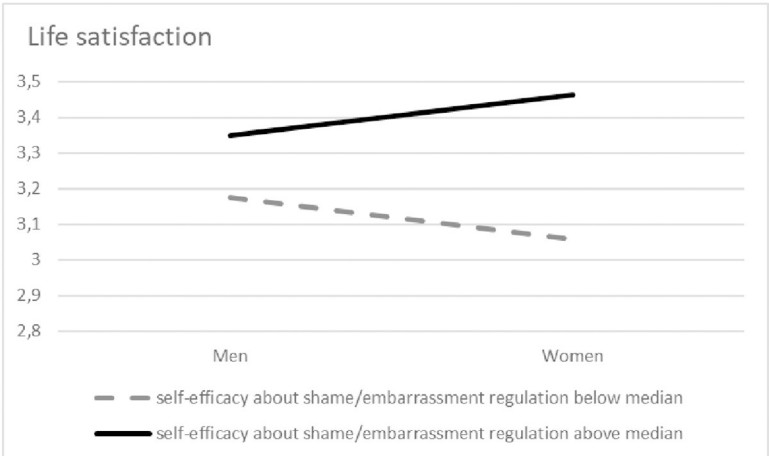

**Fig 2.** A: The effect of the interaction of sex with self-efficacy about fear regulation on life satisfaction. B: The effect of the interaction of sex with self-efficacy about shame/embarrassment regulation on life satisfaction.

common factor to all five emotional self-efficacy beliefs, since the statistical divide between the two solutions is minimal. Special attention is merited regarding the distinct assessment of the examined emotions, and their impact on the individual's well-being and social adjustment. As stated in the premises, we feel that self-efficacy beliefs may be generalized within and across domains as the expression of a common self-system that oversees the relationships of the individual with oneself and the environment. This leads to the view that emotions define a domain where one may examine how the impact of self-efficacy beliefs can be generalized across situations and tasks. Yet the extent to which the regulation of emotions represent a domain of functioning that relies upon the same mental structures and processes requires further study. In

this regard, previous findings invite us to address positive and negative emotions as expressions of different systems whose regulation depends upon different self-efficacy beliefs. Thus, we are resistant to embrace the idea of a general emotional self-efficacy, in accordance with previous criticisms of the idea of general efficacy [13].

In addressing negative emotions, we did not exclude that they could be traced to a hierarchical model. However, the present and previous findings invite us to be cautious about the extent to which the regulation of negative emotions can be traced to a common sense of mastery. Currently, the structure of self-efficacy beliefs assessed by the MNESRES parallels the common distinction between basic emotions like anger, sadness and fear, and that of self-conscious moral emotions like shame and guilt. The results of the CFA indicate they are correlated no less than they are distinctive. Thus, one may surmise that the common distinction between basic and self-conscious/moral emotions also holds well for self-efficacy beliefs in terms of the management of those emotions. It is also likely that the regulation and the beliefs of mastery of basic emotions and of self-conscious/moral emotions rely on mechanisms and strategies that differ from one to the other.

Our findings show that self-efficacy beliefs related to basic emotions are more strongly associated with negative affect and life satisfaction than self-efficacy beliefs related to self-conscious/moral emotions. This is consistent with our expectations positing the beliefs people hold about their capacity to manage negative basic emotions as better predictors of well-being than people's beliefs about their capacity to manage self-conscious/moral emotions. Thus, one might argue that self-efficacy beliefs related to basic emotions may significantly affect and, even subsume, self-efficacy beliefs related to self-conscious/moral emotions. However, further research is required to clarify this hypothesis.

In accordance with the findings from an Italian sample [38], men's self-efficacy beliefs in dealing with all the negative emotions examined were higher than women's corresponding beliefs. Although an overall significant age effect emerged, none of the specific MNESRES factors were seen to have age differences in the three groups examined. In addition, no age by gender effect emerged for such factors. In other words, self-efficacy beliefs in dealing with basic and self-conscious/moral emotions were similar among young, middle and older adults. In the Spanish context, and in agreement with studies on adult cohorts from Italy [19], Germany [15] and the United States [39], the beliefs people hold about their ability to manage negative emotions were generally negatively associated with negative affect and positively with life satisfaction. Although correlations do not allow us to establish a causal relationship, it is reasonable to surmise that being confident of one's own capacity to manage negative emotions contributes to enhance life satisfaction and to counteract negative feelings. This does not rule out that satisfaction with one's own life may contribute to being confident of one's own capacity to manage emotions. However, it is likely that the impact of self-efficacy beliefs on positive feelings and satisfaction is reciprocal. In this regard, the social cognitive theory provides guidance as how to enhance self-efficacy beliefs, which in turn can be instrumental in enhancing both positive feelings and life satisfaction.

Finally, the relationships that emerged from the regression analyses between self-efficacy beliefs related to different emotions and negative affect and life satisfaction are worthy of consideration. In general, perceiving themselves capable of managing sadness and fear proved to have a higher impact on sustaining life satisfaction and counteracting negative affect, than perceiving themselves capable of managing anger. Children probably learn how to refrain from anger earlier through socialization practices that sustain empathic concern and that lead them to avoid reactions that may hurt others. By contrast, sadness and fear are insidious experiences that are difficult to avoid and hard to handle throughout life. In this regard, the major impact

that self-efficacy beliefs associated with the management of fear has on life satisfaction and negative affect in males rather than females deserves special attention.

The results obtained represent an important contribution to the current literature on emotional regulation as they provide new information regarding the role of self-efficacy beliefs in dealing with basic and with self-conscious/moral emotions. The data corroborate the distinction of self-efficacy beliefs among discrete emotions. Likewise, the data corroborate the validity of the measures that have been developed to study the contribution of self-efficacy beliefs to emotions' regulation and their impact on different aspects of well-being, like negative affect and life satisfaction. The fact that the main effects of self-efficacy beliefs hold well across gender and age-groups is particularly noteworthy.

That said, we are aware that there are some limitations to our study. These mostly concern the overall representation, the generality of the findings and the direction of the influences among the correlated variables.

The alpha values of the MNESRES subscales for anger and sadness are far from optimal indices and this represents an important limitation. We are inclined to believe that this is due to the limited number of items, although this issue should be addressed in future studies by either enlarging the number of items or through their reformulation.

It is important to note that our sample was a convenience sample and therefore, its generalization is somewhat limited (e.g., the age groups do not match perfectly the age distribution in Spain at the time when the study was carried out). This fact may also reflect the potential dependence in terms of the social relationships among the participants. However, the fact that the major findings were replicated across both sex and gender, and that they corroborate the data from previous studies, are witness to their robustness.

As stated above, the correlations resulting from the cross-sectional design only allow us to guess about causal influences. It is likely, however, that self-efficacy beliefs dealing with emotions have an impact on people's chronic moods and life satisfaction by setting favourable conditions to learn from experience and to manage them at best.

As effective emotional regulation involves adapting strategies and deploying them effectively, studying the kinds of strategies commonly implemented by self-efficacious people to manage distinct negative emotions is an issue worthy of further study. Since previous study have shown that self-efficacious people tend to use more reappraisal than suppression strategies [15] future studies might examine whether the same patterns occur when dealing with fear, shame, embarrassment and guilt, as assessed by the MNESRES.

Finally, the results presented here highlight the importance of self-efficacy beliefs in terms of the regulation of discrete negative emotions in middle and late adulthood, in addition to what has already been established with young adults in previous studies [14, 19]. These results may set the basis to further investigate the degree to which the strengthening of emotional self-efficacy beliefs can represent a viable strategy to equip people to face major life transitions and successful aging.

## Supporting information

**S1 Data. Data regarding Tables 1, 2, 3, 4 and 5.**
(XLSX)

**S2 Data. Data regarding Fig 1.**
(XLSX)

**S3 Data. Data corresponding to Fig 2.**
(XLSX)

**S1 Models.**
(PPTX)

## Author Contributions

**Conceptualization:** Mariagiovanna Caprara, José Bermúdez, Gian Vittorio Caprara.

**Data curation:** José Bermúdez.

**Methodology:** Laura Di Giunta.

**Supervision:** Mariagiovanna Caprara, Gian Vittorio Caprara.

**Writing – original draft:** Mariagiovanna Caprara.

**Writing – review & editing:** José Bermúdez.

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
