## [Decision Letter · Decision Letter 0]

15 Apr 2020

PONE-D-20-00499

How Mastery Beliefs in Dealing with Negative Emotions Can Counteract Negative Affect and Promote Life Satisfaction Across Gender and Age

PLOS ONE

Dear Mrs Caprara,

Thank you for submitting your manuscript to PLOS ONE. After careful consideration, we feel that it has merit but does not fully meet PLOS ONE’s publication criteria as it currently stands. Therefore, we invite you to submit a revised version of the manuscript that addresses the points raised during the review process.

It has been particularly difficult to find reviewers, which may account for the unusual long time between the submission and the decision. 

We would appreciate receiving your revised manuscript by May 30 2020 11:59PM. To enhance the reproducibility of your results, we recommend that if applicable you deposit your laboratory protocols in protocols.io, where a protocol can be assigned its own identifier (DOI) such that it can be cited independently in the future. For instructions see: http://journals.plos.org/plosone/s/submission-guidelines#loc-laboratory-protocols

We look forward to receiving your revised manuscript.

Kind regards,

Delphine Grynberg, PhD

Academic Editor

PLOS ONE

Reviewers' comments:

Reviewer's Responses to Questions

**Comments to the Author**

1. Is the manuscript technically sound, and do the data support the conclusions?

Reviewer #1: Yes

Reviewer #2: Partly

Reviewer #3: Yes

2. Has the statistical analysis been performed appropriately and rigorously? 

Reviewer #1: Yes

Reviewer #2: I Don't Know

Reviewer #3: Yes

3. Have the authors made all data underlying the findings in their manuscript fully available?

Reviewer #1: Yes

Reviewer #2: Yes

Reviewer #3: Yes

4. Is the manuscript presented in an intelligible fashion and written in standard English?

Reviewer #1: Yes

Reviewer #2: No

Reviewer #3: Yes

5. Review Comments to the Author

Reviewer #1: The manuscript deals with the question of the factorial structure and construct validity of the Multidimensional Negative Emotions Self-Regulatory Efficacy Scale (MNESRES), which measures self-efficacy beliefs in regulating discrete negative emotions. Overall, the manuscript seems scientifically rigorous to me. However, I have some questions and comments.

(1) Clear presentation of the aims, methods, & results

(a) Throughout the manuscript, the writing was not always very clear / concise.

(b) The introduction was too broad for me. The MNESRES, which should be front and center, is only mentioned at the end of the second page. I would have expected an introduction that focusses more on why this particular study is of importance. I would also have liked to have more information on why self-regulatory efficacy may be comparable across age groups and gender. Also, the rationale for choosing the two outcomes (negative affect & satisfaction with life) was not clear to me.

(c) Given that the authors stress the importance of shame and guilt, it seemed somewhat surprising to me that they expected stronger associations with negative affect (NA) and satisfaction with life (SWL) for the basic emotions.

(2) Methods

(a) Sample – The sample consisted of relatives and friends of undergraduate Psychology students. The authors may want to discuss that this is not ideal, given that participants were not independent from one another (e.g., couples, relatives, friends, etc.). Also, middle-aged men were oversampled and the distribution of age in the older age group seems skewed.

(b) MNESRES – I would help my understanding of the concept of self-regulatory efficacy to have a table with the MNESRES items or at least example items for each dimension.

(c) Negative affect – I was wondering why only NA was reported, but not PA. Given that the authors also assessed SWL, it would have made sense to report on subjective well-being (with the facets NA and PA for affective well-being, and SWL as the cognitive facet; see Diener). Alternatively, it would have been interesting to have measures of discrete emotions matched to the dimensions of the MNESRES as outcomes.

(3) Analytic approach

(a) It was unclear to me why the three proposed models were tested. How were these models pre-selected?

(b) Some authors have argued for CFI values to be close to .95 rather than .90 (Hu & Bentler, 1999). To me, .90 seems like a rather low bar – maybe the authors want to discuss this.

(c) It was not clear to me, why the authors did not test three-way interactions (MNESRES x age x sex).

(d) It was not clear to me why the authors tested the different linear regression models the way they did. It seemed to me that they wanted to show that although some of the MNESRES facets did not have predictive value above and beyond the other facets, there were some bivariate associations. Two options occur to me: (1) reporting a bivariate correlation table (which I would suggest anyway), and/or (2) reporting regression coefficients for each individual MNESRES facet (controlling for age and sex in each model) & for all combined facets. Also, I would like to see the full models and not just the interaction effects in Step 3. The presentation of results in Table 5 seems slightly misleading to me as the effect of SE_SE of course turns insignificant when adding the predictors in Step 3 (third set of HRA).

(4) Results

(a) CFA model comparisons: Please note that I am not an expert on SEM. I did not understand why the authors chose Model 2 (correlated factors) instead of a model with a superordinate factor. I see that the AIC was slightly lower in the former model. However, on theoretical grounds, it seems to me that the assumption is that the different factors are part of one underlying construct, namely, self-regulatory efficacy in managing negative emotions. It seems so much more parsimonious when predicting other outcomes to only have one score (i.e., the superordinate score); in particular, when the outcome is as broad as NA (i.e. not at the same level of discrete emotions) or SWL. I would be curious how reliable and predictive an aggregated score would be.

(b) Interaction effects in the prediction of SWL (regression): The interpretation of the interaction effects does not seem quite right to me. The interaction effects show differences between male and female participants. I am not sure how the data was coded, but assuming that men = 0 and women = 1, self-efficacy for regulating fear was more strongly associated with SWL for men than for women, and the other way around for shame. Please check and report the coding in the manuscript.

(5) Discussion

(a) The authors state that they found support for the reliability and validity of the scale. However, while the authors investigate the factorial structure, the reliability of the measure was not tested (in terms of internal consistency or re-test reliability). In terms of validity, the authors tested the predictive validity, but not convergent and discriminant validity, for example. Therefore, the authors may want to be more specific with their conclusions.

(b) I do not understand why the authors conclude that “one may surmise that the common distinction between basic and self-conscious/moral emotions holds good also for self-efficacy beliefs relating to the management of those emotions.” (p. 27). If that was the case, shouldn’t the factor structure be different (i.e., two factors)?

(c) I would have wanted the authors to discuss the fact that guilt and shame had no predictive value above and beyond the other facets. One hypothesis that came to my mind was that these emotions may be relatively rare in healthy adults’ everyday life?

(6) Other comments

(a) Sometimes, sign. results are presented in bold print in the tables, and sometimes not. In Table 4, the estimate for the SE_G X Sex interaction is printed in bold although it is not sign (outcome SWL).

(b) I was not entirely sure whether mastery beliefs and self-efficacy were used synonymous here.

(c) The data (excel) is not sufficiently labeled. Also, labeling in English would be appreciated.

Reviewer #2: The researchers addressed three aims: (1) examine measurement invariance of the MNESRES across gender and age in a Spanish sample; (2) examine gender and age differences in self-regulatory self-efficacy beliefs using the MNESRES; and (3) examine associations between discrete emotion self-regulatory self-efficacy beliefs, negative affect, and life satisfaction. These are interesting and important aims and I commend the authors for this study. However, I think several aspects of the presentation and analyses could be improved. I have outlined my specific concerns and suggestions below.

General Writing:

1. The clarity and organization of the manuscript could be improved. In general, the sentences are very long and the paragraphs are rather short. This made it difficult for me to follow the authors' logic from one idea to the next in many locations.

2. There are grammatical errors and inconsistencies throughout the manuscript that should be addressed. For example, individual's is used where individuals' should be used instead in multiple locations. Hyphens in words like life-satisfaction, age-group, and "life-experience" are unnecessary and used inconsistently.

3. In the abstract and in several locations throughout the manuscript, the authors combine all three research aims into a single sentence or phrase. This makes it difficult to understand the key research questions. Instead, the authors should consider stating the three aims separately.

4. The authors should remove the causal language from the Title ("counteract" and "promote").

5. It is unclear throughout whether the basic emotions versus moral emotions distinction is a key research question/guiding motivation of the present research or if it is a secondary exploratory analysis.

6. The manuscript should be carefully proofread and edited for clarity.

Introduction:

7. The first sentence of the Introduction is much broader than the topic of the paper. An opening sentence focused on self-efficacy beliefs (rather than theories of personality more generally) would be more appropriate.

8. The Introduction could benefit from clearer organization, perhaps with subheadings or at least specific paragraphs focused on the different components of the research question (e.g., self-efficacy beliefs, the structure of emotion regulation self-efficacy beliefs, the role of discrete emotions, associations between self-efficacy beliefs and well-being outcomes).

9. Several assertions in the Introduction lack references. (e.g., "While people display basic emotions like anger and sadness from the early stages of life, self - conscious/moral emotions do not appear unless self-structures and processes develop..." and "A large body of research has documented that the beliefs people hold about their capacity to cope efficaciously with specific challenges and to face demanding situations, exert a pervasive influence over thought, motivation and action across different domains of functioning.").

10. The Introduction focused heavily on scale development and is missing relevant literature to links between emotion regulation self-efficacy beliefs and well-being outcomes. This seems like an important literature to review, given that it is a key focus of the present research. Relatedly, the authors should consider reviewing emotion regulation self-efficacy research beyond the MNESRES (e.g., Emotion Regulation Questionnaire - Self Efficacy; Goldin et al., 2009; beliefs about emotions; Ford & Gross, 2019).

11. The following hypothesis seems much too broad to come from the reviewed evidence: “These findings further corroborate the hypothesis postulating that distinct discrete emotions play a different role with regard to adaptation and, as such, they require a specific consideration regarding their assessment and their treatment.” Is this the authors' hypothesis? If so, it should be significantly narrowed. If it is an existing hypothesis from the literature, a reference should be cited and it should be used to support the researchers' focus on discrete emotions rather than the other way around. In other words, this broad hypothesis can motivate the researchers' more specific research question, however, the researchers' specific research question cannot motivate this broader hypothesis.

Method and Results

12. What does "Within such not previously explored age group," refer to? In the next paragraph, the authors note that the scale was previously validated in an Italian sample of young to older adults.

13. The researchers should list all 10 negative emotional states in the PANAS. This is crucial information for interpreting findings given that the research question is about discrete emotions. Readers shouldn’t have to refer to a separate reference to obtain this information. My key question here is, are basic emotions and self-conscious emotions both represented in the PANAS items? Are the five discrete emotions from the MNESRES represented in the PANAS?

14. The authors need to justify their analytic approach, in particular, the hierarchical regressions.

14a. Why were age AND gender interactions included together? This makes for a more complex model that could obscure significant interactions.

14b. Why were moral emotions and basic emotions included in different steps. Was this based on theory? Was this based on the observed correlations or the result of the first set of regressions? If so, this should be noted.

15. Given the focus on basic versus moral emotions, I am surprised the researchers did not test this distinction in the factor analysis. Was there a reason for not testing a model with a basic emotions factor and a moral emotions factor?

16. The abbreviations for the different discrete emotions are hard to remember. The authors should consider using more descriptive abbreviations or the complete terms (e.g. SE_fear).

Discussion:

17. I was surprised that the authors did not discuss the differential associations of discrete emotions versus moral emotions with well-being, given the focus on this in the Introduction and Results. Could these results be explained simply by the fact that basic emotions occur more frequently and/or are better represented in the PANAS?

Reviewer #3: Review Comments to the Author

Please use the space provided to explain your answers to the questions above. You may also include additional comments for the author, including concerns about dual publication, research ethics, or publication ethics. (Please upload your review as an attachment if it exceeds 20,000 characters) (Limit 200 to 20000 Characters)

Could you please see my attached review, I have added all my comments to the Author in an attachment.

6. PLOS authors have the option to publish the peer review history of their article (what does this mean?). If published, this will include your full peer review and any attached files.

Reviewer #1: No

Reviewer #2: No

Reviewer #3: No

---

## [Author Response · Author response to Decision Letter 0]

8 Sep 2020

We have incorporated all the reviewers'suggestions into our revision. They were very helpful. Thank you

---

## [Decision Letter · Decision Letter 1]

8 Oct 2020

PONE-D-20-00499R1

How Self-efficacy Beliefs in Dealing with Negative Emotions are associated to Negative Affect and to Life Satisfaction across Gender and Age

PLOS ONE

Dear Dr. Caprara, 

Thank you for submitting your manuscript to PLOS ONE. After careful consideration, we feel that it has merit but does not fully meet PLOS ONE’s publication criteria as it currently stands. Therefore, we invite you to submit a revised version of the manuscript that addresses the points raised during the review process.

Please submit your revised manuscript by 28th October. If you will need more time than this to complete your revisions, please reply to this message or contact the journal office at plosone@plos.org. Please include the following items when submitting your revised manuscript:

We look forward to receiving your revised manuscript.

Kind regards,

Paola Iannello

Academic Editor

PLOS ONE

Reviewers' comments:

Reviewer's Responses to Questions

**Comments to the Author**

1. If the authors have adequately addressed your comments raised in a previous round of review and you feel that this manuscript is now acceptable for publication, you may indicate that here to bypass the “Comments to the Author” section, enter your conflict of interest statement in the “Confidential to Editor” section, and submit your "Accept" recommendation.

Reviewer #1: (No Response)

Reviewer #2: (No Response)

Reviewer #3: All comments have been addressed

2. Is the manuscript technically sound, and do the data support the conclusions?

Reviewer #1: Partly

Reviewer #2: Yes

Reviewer #3: Yes

3. Has the statistical analysis been performed appropriately and rigorously? 

Reviewer #1: Yes

Reviewer #2: Yes

Reviewer #3: Yes

4. Have the authors made all data underlying the findings in their manuscript fully available?

Reviewer #1: Yes

Reviewer #2: Yes

Reviewer #3: No

5. Is the manuscript presented in an intelligible fashion and written in standard English?

Reviewer #1: Yes

Reviewer #2: Yes

Reviewer #3: Yes

6. Review Comments to the Author

Reviewer #1: I read the manuscript for the second time. The revised version is much more concise and much easier to follow. I also very much appreciate that the authors now provided the data with the relevant variables labeled in English. However, I still have a few concerns.

(1) Regulation of basic emotions vs. moral emotions: I still had some difficulty following the argument that the regulation of basic emotions should be more important than the regulation of moral emotions. For parts of the argument, references to relevant research findings seem to be missing. For example, the authors write (pp. 5-6): “Indeed, negative basic emotions like anger, fear and sadness have a pervasive impact over individuals’ life, and their regulation has a primary influence on adaptation when compared to self-conscious/moral emotions like guilt and shame.” I think such a statement (and similar statements in the manuscript) would need a reference.

(2) Reliabilities of the MNESRES subscales were low for anger and sadness (.55, .54). This could be discussed.

(3) Figure 1: Some correlations between factors seem to be missing from the figure (e.g., correlations between SE_G and SE_DS.

(4) Interaction effects/ Figure 2: I still think that the authors’ interpretation of the interaction effects is (a) not really in line with their research question and (b) not 100% correct. With regard to (a), I think what the authors did (and illustrated), is considering the SEs as moderators of the association between sex and satisfaction with life. However, since the research is concerned with associations between the SEs and satisfaction with life, it makes much more sense to me to consider sex as the moderator here. Statistically, there is of course no difference between considering sex or the SEs as moderators. However, the interpretation is slightly different. The results show that there is an association between SE_fear and life satisfaction: Individuals with higher SE_fear have higher life satisfaction. Treating sex as a moderator, the interaction effect (SE_fear x sex, b= -.12) means that for women, the association (semi-partial correlation) between SE_fear and life satisfaction is weaker than for men. Although there is no main effect of SE_shame on life satisfaction, there is an interaction effect SE_shame x sex (b=.14), indicating that for women, SE_shame is related to satisfaction with life, but not for men.

With regard to (b), the authors write: “life satisfaction was particularly higher in women with high SE_Shame than in both men and women with low SE_Shame (Figure 2B), as well as in men with high SE_Shame.” This cannot be concluded from the interaction effect, which only indicates that there are differences in trajectories. One would have to test these specific differences between higher and lower groups post-hoc. In the plots presented in the paper, it is fairly obvious that women high in SE_shame are not more satisfied with their life than men with high SE_shame (the difference is only about 0.1 points on the scale). As sex is a categorical variable here, I also do not think that plotting the data as trajectories in the way presented in the paper is most useful. Therefore, I would strongly suggest to switch to considering sex as the moderator, and changing the interpretation and the plots accordingly.

(5) Discussion: I think that some statements made by the authors in the discussion need further elaboration. I also felt that some conclusions go beyond what was studied.

(a) “Albeit with caution, it may be concluded that the five-oblique-factor solution further corroborates the view that self-efficacy beliefs are associated with the regulation of discrete emotions.” (p.20) Since the questionnaire measures self-efficacy beliefs with regard to regulating discrete emotions, I am not sure how a factor analysis can corroborate the association between self-efficacy beliefs and regulation. Maybe the authors could rephrase?

(b) “However, the present and previous findings invite us to be cautious about the extent to which the regulation of negative emotions can be traced to a common sense of mastery.” (p. 21) – In the present study, a model with one common mastery factor was not tested. The fact that a model with five correlated factors fit the data best does very much invite the idea that there may be a common factor, I think. I still think that the best option here would be to simply test a model with a common factor. As a reader of this paper, I would be very interested in that.

(c) “Thus, one may surmise that the common distinction between basic and self-conscious/moral emotions also holds well for self-efficacy beliefs in terms of the management of those emotions.” (p.21). The reply given by the authors regarding my question in the first review does not convince me. I still fail to see how the present study supports this conclusion. Based on the CFAs, a model with 5 correlated factors fit the data best, which is a model that does not support this distinction. I understand that the fit of alternative models was also not that bad, but I do not think that this is strong evidence to corroborate a distinction. In fact, the correlations between the subscales (Table 4) show that the correlations between SE_shame with SE_sadness and with SE_fear were similar to the correlation between SE_anger and SE_fear, which does not indicate that SE_shame belongs to a different factor than the other SEs. Furthermore, all factors correlated with the outcomes (negative affect and satisfaction with life). The fact that SE_shame and SE_guilt were not uniquely predictive of the outcomes above and beyond all other factors is not indicative of these factors not belonging to the same underlying construct. It may very well be the case that the distinction makes sense conceptually and that this was found in other studies, but in the present study, the data does not support such a conclusion, I would say.

(d) “One might think that most people learn how to refrain from anger earlier” (p.22) – Why might one think that? (see Comment #1

Reviewer #2: The authors addressed the majority of comments raised in the prior round of revision. I appreciated that the authors tempered their conclusions about the factor structure and direction of causality in the Discussion section. I also appreciated that they considered both evolutionary and constructivist views of emotion in the Introduction. Overall, this is a thorough revision of a nice manuscript. I have outlined a small number of minor suggestions below.

1. The rationale for the hypothesis that regulation of basic emotions should be more important for well-being (bottom of page 5, top of page 6) was not clear.

2. Please name all 10 emotions that were included in the negative emotion measure. This information is important for evaluating the degree of overlap between the self-efficacy beliefs measure and the negative emotion measure.

3. It is unclear what is meant by this sentence in the Discussion: "Since the sample was taken mostly from a relatively affluent middle class population, one cannot exclude significant variations due to age and gender, either in the means or in the relationship of the variables examined, particularly among a population whose conditions of life are quite different." I believe the authors are trying to make a point about lack of generalizability beyond middle class population, but it is unclear to me what they are saying about age and gender.

4. The first paragraph includes jargon and complex sentence structures. This may be personal preference and the authors may feel free to leave this paragraph as is, however I found it difficult to understand.

Reviewer #3: The authors have appropriately addressed reviewer comments, and the manuscript is now much stronger. I have no further revisions.

7. PLOS authors have the option to publish the peer review history of their article (what does this mean?). If published, this will include your full peer review and any attached files.

Reviewer #1: No

Reviewer #2: No

Reviewer #3: No

---

## [Author Response · Author response to Decision Letter 1]

29 Oct 2020

We are grateful for the constructive comments of the reviewers, which we believe have helped us to present our data more clearly. In the attached cover letter we detail our responses to the reviewers’ comments and we indicate the changes made in the revised manuscript.

---

## [Editor Report · Decision Letter 2]

2 Nov 2020

How Self-efficacy Beliefs in Dealing with Negative Emotions are associated to Negative Affect and to Life Satisfaction across Gender and Age

PONE-D-20-00499R2

Dear Dr. Caprara,

We’re pleased to inform you that your manuscript has been judged scientifically suitable for publication and will be formally accepted for publication once it meets all outstanding technical requirements.

Kind regards,

Paola Iannello

Academic Editor

PLOS ONE
---

## [Editor Report · Acceptance letter]

9 Nov 2020

PONE-D-20-00499R2 

How Self-efficacy Beliefs in Dealing with Negative Emotions are associated to Negative Affect and to Life Satisfaction across Gender and Age 

Dear Dr. Caprara:

I'm pleased to inform you that your manuscript has been deemed suitable for publication in PLOS ONE. Congratulations! Your manuscript is now with our production department. 

Kind regards, 

on behalf of

Dr. Paola Iannello 

Academic Editor

PLOS ONE